# Enhancing the Thermal Stability and Reducing the Resistance Drift of Sb Phase Change Films by Adding In₂Se₃ Interlayers

**Feng Su** [1], **Yifeng Hu** [1,2,3,*], **Xiaoqin Zhu** [1] and **Tianshu Lai** [4]

1   School of Mathematics and Physics, Jiangsu University of Technology, Changzhou 213000, China; 15189781173@163.com (F.S.); pcram@jsut.edu.cn (X.Z.)
2   State Key Laboratory of Silicon Materials, Zhejiang University, Hangzhou 310027, China
3   Engineering Research Center of Digital Imaging and Display, Ministry of Education, Soochow University, Suzhou 215006, China
4   State-Key Laboratory of Optoelectronic Materials and Technology, School of Physics, Sun Yat-Sen University, Guangzhou 510275, China; stslts@mail.sysu.edu.cn
*   Correspondence: hyf@jsut.edu.cn

**Abstract:** In this paper, pure Sb and composite multilayer $In_2Se_3$/Sb thin films were prepared on a $SiO_2$/Si substrate. The effects of the addition of $In_2Se_3$ interlayers on the physical and electrical properties of phase change thin films were investigated. Compared with pure Sb film, the composite multilayer $In_2Se_3$/Sb film had a higher crystallization temperature (~145 °C), larger crystallization activation energy (~2.48 eV), less resistance drift (~0.0238) and better thermal stability. The results of X-ray photoelectron spectroscopy indicated that the In-Sb bond was formed in the multilayer $In_2Se_3$/Sb film. The near infrared spectrophotometer showed that the band gap changed at different annealing temperatures. Changing the annealing temperature of the film allowed for the phase structure of the film to be studied by using X-ray diffractometer. The surface morphology and electrostatic potential at different annealing temperatures were using atomic force microscope. It was found that the flat film had a smoother surface. Phase-change memory devices based on $[In_2Se_3(4\ nm)/Sb(6\ nm)]_8$ film reduced power consumption by approximately 74% compared to pure Sb film. In conclusion, the $In_2Se_3$ interlayers effectively inhibited the resistance drift of the phase change thin film and enhanced its thermal stability.

**Keywords:** $In_2Se_3$ interlayers; resistance drift; thermal stability; phase change memory



## 1. Introduction

Phase change semiconductor memory refers to sulfur compound random access memory (C-RAM). The superiority of C-RAM is fully demonstrated only when the size of the device unit reaches the three-dimensional nanoscale, thus it developed very slowly for a long time. With industrial preparation techniques and processes reaching deep submicron and even nanometer scales, the size of the phase change materials in devices can be reduced to the order of nanometers. The voltage and power required for the materials to undergo the phase transition are greatly reduced and can be matched with existing Complementary Metal-Oxide-Semiconductor (CMOS) technology. Since then, phase change memory technology has entered a rapid development stage. In recent years, non-volatile storage technology has made a series of significant advances in many aspects, which brings new opportunities for improving the storage energy efficiency of computer systems [1]. As one of the various types of semiconductor memories, phase change memory (PCM) is a kind of nonvolatile memory [2]. It stores data using the electrical conductivity difference between crystalline and amorphous states, and it consequently possesses great research value [3]. Due to the low crystallization temperature of Sb, its amorphous thermal stability is poor, which limits its use in commercial applications [4]. Researchers have chosen to develop phase change materials with rapid crystallization rates,

high transformation temperatures, good stability and low energy consumption. Doping is a common method used to develop new materials, which is certain to significantly improve their properties. However, the use of doping often increases the diversity of components and power consumption, slows down the running speed of storage devices, and reduces the precision control of components while improving thermal stability, which is not conducive to the current requirements of high-density and large-scale integration [5]. $Ge_2Sb_2Te_5$ (GST) is a widely studied phase change material. However, it has encountered some obstacles in its development, such as poor amorphous thermal stability, resistance drift and higher power consumption. The element Te also has certain toxicity, which pollutes the semiconductor process and causes certain pollution to the environment. It is urgent to develop new phase change materials with excellent properties.

In this paper, the multilayer composite method is adopted by adding an $In_2Se_3$ (IS) interlayer to Sb film [6,7]. It can give full play to the original properties of each element without destroying the original structure of the phase change material [8]. At the same time, the synergistic and complementary advantage effect can be achieved through the coupling effect between the two kinds of film, so as to improve the performance of the materials [9]. Magnetron sputtering technology is adopted to prepare Sb and $In_2Se_3$/Sb (IS/Sb) composite films on a $SiO_2$/Si substrate. The application of the IS/Sb multilayer phase change film in PCM was evaluated by testing the properties of the materials and devices. The results show that the composite multilayer IS/Sb phase change film has a higher crystallization temperature and crystallization activation energy. The IS interlayers effectively inhibit the resistance drift of the phase change film and enhance its thermal stability. Compared with Sb phase change materials, IS/Sb phase change thin film has certain advantages and will have broad application prospects in the field of phase change memory.

## 2. Experimental

RF magnetron sputtering technology was adopted to prepare Sb and IS/Sb composite film on a $SiO_2$/Si substrate. According to the sputtering rate of the target, the sputtering time was set to control the thickness of the film. In the experiment, we used two targets with a sputtering power of 30 W. Argon ions were injected into the sputtering process and the argon flow rate was set to 30 SCCM. The sputtering pressure was 0.4 Pa and the pressure of the vacuum chamber was $4 \times 10^{-4}$ Pa. The uniform depositing of the film on the silicon substrate was ensured by keeping the sample plate rotating at a rate of 20 r/min during the sputtering.

The resistance-temperature curves of the film were measured using in situ film resistance measurement. The crystallization-time curves and crystallization activation energy of the films were measured. The reflectance spectrum of the thin film was observed using a near infrared spectrophotometer with wavelength range of 400–2500 nm. X-ray diffractometer (XRD) was used to test the X-ray lines of the film between 10° and 60°. The resistance drift of the film at 60 °C was observed. The bonding state was observed using X-ray photoelectron spectroscopy (XPS). The appearance and electrostatic potential of the film were observed under atomic force microscope (AFM) in semi-contact mode. A self-made PCM device was produced, and its electrical performance was tested.

## 3. Results and Discussions

Figure 1a shows the resistance curves of Sb and IS/Sb film with different components, and it is heated at a rate of 20 °C per minute. To reduce the effect of total thickness and period number, the same total thickness of 50 nm and period number of 5 were used for all the films. As can be seen from Figure 1a, all the films maintained high initial resistances, indicating that the deposited state was amorphous. As the temperature rose, their resistances began to decline slowly, mainly due to the negative resistance temperature characteristic of the semiconductor. When the temperature rose to $T_c$, both the phase transition temperature [10] and the resistances of the films began to drop sharply, and

the amorphous film converted to an ordered crystalline state [11]. The $T_c$ of pure Sb is 107 °C, and the $T_c$ was gradually increased by adding multiple In$_2$Se$_3$ interlayers. When the thickness ratio of In$_2$Se$_3$ to Sb reaches 4:6, its $T_c$ rises to 145 °C, and its thermal stability is enhanced significantly. The amorphous resistance increases from about $7 \times 10^4$ $\Omega$ to about $5.3 \times 10^5$ $\Omega$, and the crystalline resistance increases from 251 to 2670 $\Omega$. In light of the Joule heat formula $Q = I^2 \cdot R \cdot t$, in a self-heating system utilizing the thermal effect of current, the increase in resistance helps to reduce the power consumption of phase change film during reversible transformation operations.

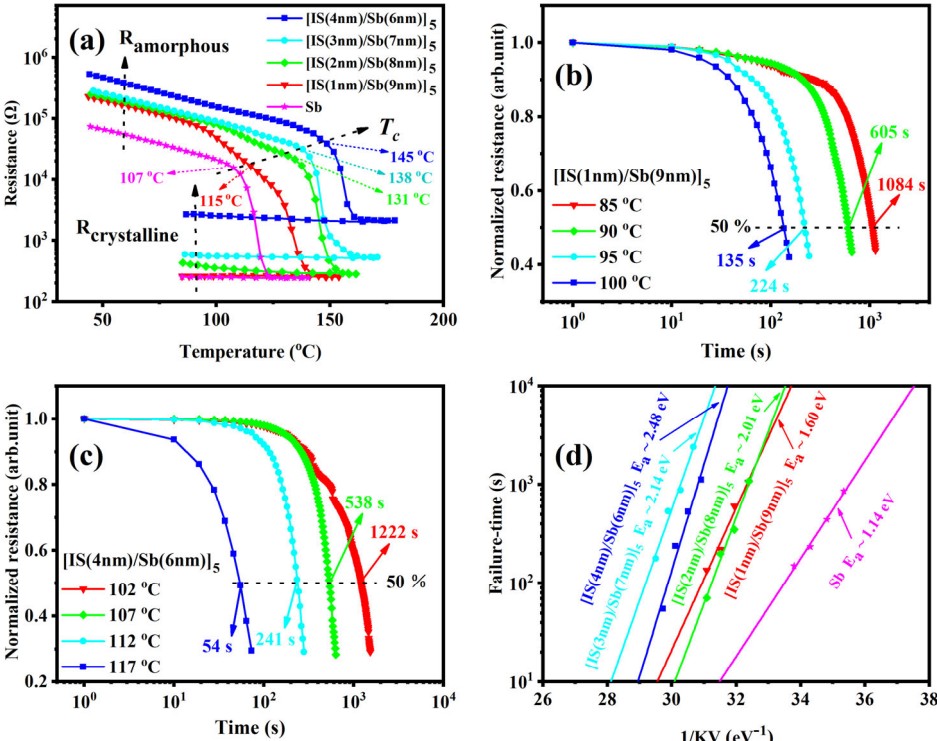

**Figure 1.** (**a**) The resistance film of temperature change at a rate of 20 °C/min; (**b**) Normalized resistance curves with time of [IS(1 nm)/Sb(9 nm)]$_5$ films after isothermal annealing at 100, 95, 90 and 85 °C respectively; (**c**) Normalized resistance curves with time of [IS(4 nm)/Sb(6 nm)]$_5$ films after isothermal annealing at 117, 112, 107 and 102 °C, respectively; (**d**) Crystallization activation energies $E_a$ of Sb and IS/Sb composite films.

Figure 1b,c show the isothermal crystallization curves of two films (other components are not shown here), and their isothermal temperatures are all lower than their $T_c$. With the increase in holding time, phase transition occurs when the film accumulates to a certain amount of energy [12]. Due to the division of the crystallization process of the film into four stages, namely the incubation period, the nucleation period, the growth period and the grain coarse period, the crystallization curve of the film presents a slow and then steep decline process. We define the failure time as the time when the resistance drops to 1/2 of the initial value [13]. The isothermal crystallization experiments of [IS(1 nm)/Sb(9 nm)]$_5$ are conducted at 85, 90, 95 and 100 °C, respectively. It can be seen that the failure time of the film increases significantly from 135 s to 1084 s with the decrease in isothermal temperature, indicating that more time is needed to cumulate enough energy for the phase change at a lower isothermal temperature. Figure 1c,b have similar shapes, but Figure 1c conducts isothermal crystallization experiments at 102, 107, 112 and 117 °C, respectively, mainly because [IS(4 nm)/Sb(6 nm)]$_5$ films have a higher $T_c$ (~145 °C). Correspondingly, the failure times are 1222, 538, 241 and 54 s, respectively. The failure times and isothermal

temperatures of all the films were fitted, and Figure 1d was obtained. According to the Arrhenius equation [14,15]:

$$t = \tau_0 \exp\left(\frac{E_a}{k_b T}\right) \qquad (1)$$

where $t$, $\tau_0$, $k_b$, $E_a$ and $T$ are failure time, proportional time constant, Boltzmann constant, crystallization activation energy and isothermal temperature, respectively [16]. The $E_a$ (slope of fitted lines) for all films are 1.14, 1.60, 2.01, 2.14 and 2.48 eV respectively [17]. Obviously, the increase of the thickness of the $In_2Se_3$ interlayer prevents the directional movement of carriers and increases the barrier of crystallization, which makes the film have better thermal stability [18].

Figure 2 shows the variation of the absorption coefficient of the [IS(4 nm)/Sb(6 nm)]$_{20}$ film at distinct annealing temperatures. Near-infrared spectrophotometer and the Kubelka-Munk function are used to evaluate the band gap $E_g$ of the materials [19]:

$$K/S = (1 - R)^2/(2R) \qquad (2)$$

where $K$, $S$ and $R$ are the absorption coefficient, the scattering coefficient and reflectance, respectively. The linear part of the curve is fitted and extended to the energy axis. The $E_g$ of the [IS(4 nm)/Sb(6 nm)]$_{20}$ film decreases monotonically when the annealing temperature gradually increases. Figure 2 shows that $E_g$ gradually decreases from 0.935 eV~25 °C to 0.887 eV~102 °C, 0.87 eV~122 °C and 0.861 eV~180 °C. It has been reported that the carrier density in semiconductors is proportional to $[-E_g/(2kT)]$. The band gap decreases, then the carrier will increase [12,20]. This is also the reason why the resistivity of the sample declines when the annealing temperature gradually increases. It is in accord with the *R-T* curves in Figure 1a. In addition, two linear parts of the absorption curve in Figure 2 are fitted respectively, and two slope values slop N and slop M are obtained [21,22]. The slop M value gradually increases from 28.462 at 25 °C to 31.131 at 180 °C. An increase in slope M is associated with a decrease in the randomness of the atomic configuration [23]. When the annealing temperature increases, the atomic order degree of the film increases too. Slop N also increases from 5.064 at 25 °C to 6.018 at 180 °C. The increase of slope N is caused by the decrease of the localized state of the tail band inside the material, indicating that the amorphous proportion of the material continues to decline with the increase of annealing temperature [23].

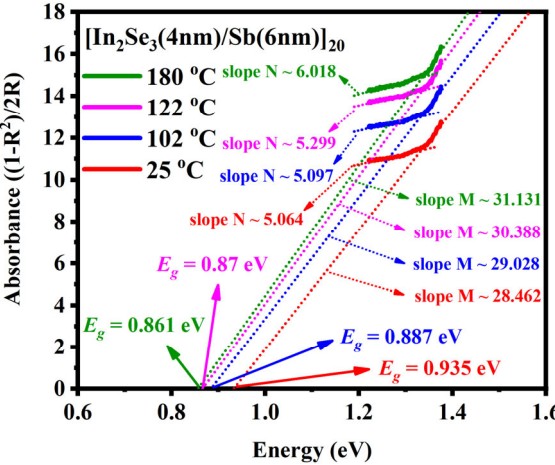

**Figure 2.** Kubelka-Munk function curves of [IS(4 nm)/Sb(6 nm)]$_{20}$ film at distinct annealing temperatures.

Figure 3a–d show the crystallization proportion-time curves of different films under isothermal annealing. The crystallization ratio curves in Figure 3a–d all show an inverse S shape, which is mainly related to the incubation period, nucleation period and growth period of the internal crystals. The incubation period and nucleation period are longer, but

the growth period is faster. Compared with Figure 3a, this stage becomes more obvious in Figure 3b–d, indicating that there are already a large number of microcrystals in the deposited Sb film. A long nucleation process is not required, which helps to obtain an ultra-fast phase transition rate. Compared with the pure Sb film, the multilayer film after composite has a longer crystallization time and a slower phase transition due to the existence of multiple interfaces. Figure 3e–h show the curves of Ln[−ln(1 − χ)]-Ln(t) [24]. The derivation of the Avrami equation assumes that the nucleation is uniform, the nucleation rate and growth rate are constant, and the nucleation time is very short. On this basis, Avrami gave the universal equation of crystallization kinetics considering the correlation between nucleation rate and time [22]:

$$\chi(t) = 1 - exp[-(Kt)^n] \tag{3}$$

where the Avrami index $n$ is related to the phase transition mechanism, and is determined by the decay of recrystallization nucleation rate. $K$ is constant and $t$ is time. Figure 3e–h show that the $n$ of the film increases from 1.560 of Sb to 1.901 of [IS(4 nm)/Sb(6 nm)]$_5$, with values less than 2.5. It indicates that their crystallizations belong to the one-dimensional crystallization mechanism, which usually corresponds to a faster phase transition rate. This can be caused by the existence of lots of Sb grains in the IS/Sb multilayer composite film and the low thermal conductivity of the multilayer structure.

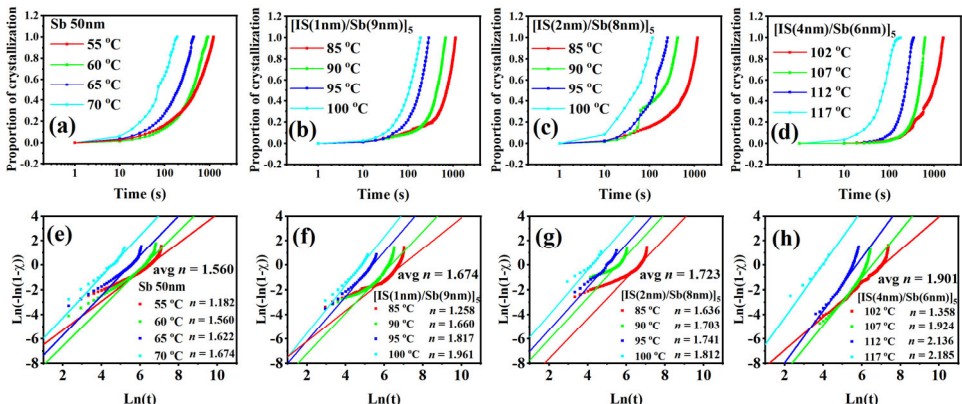

**Figure 3.** (**a–d**) Crystallization proportion-time curves of different films under isothermal annealing; (**e–h**) Ln[−ln(1 − χ)]-Ln(t) curves of different films.

Figure 4 is the XRD pattern of the [IS(4 nm)/Sb(6 nm)]$_{20}$ film after annealing at different temperatures for 5 min. There is no diffraction peak in the sedimentary state, indicating its amorphous structure. When the annealing temperature rises to 102 °C, the diffraction peak of 003 belonging to Sb appears, indicating that Sb has weak thermal stability. As the annealing temperature gradually increases, the diffraction peak 006, also belonging to Sb, appears and gradually strengthens. The Scherrer formula is used to calculate the grain size of the film after crystallization, and the expression is as follows [25]:

$$D = 0.943\lambda(\beta\cos\theta) \tag{4}$$

where $\lambda$ is the X-ray wavelength, $\beta$ is the peak width at the semi-maximum, and $\theta$ is the diffraction angle [26]. The grain size of the [IS(4 nm)/Sb(6 nm)]$_{20}$ film after annealing at 142 °C is 7.806 nm. It indicates that the grain size of the [IS(4 nm)/Sb(6 nm)]$_{20}$ film is small. In addition, the In$_2$Se$_3$ phase is not found in all the spectral lines, indicating that it exists in amorphous form. The existence of a great quantity of amorphous interlayers will reduce the crystalline region of Sb and inhibit its crystallization, which helps to enhance its thermal stability. It is consistent with the conclusion of Figure 1a.

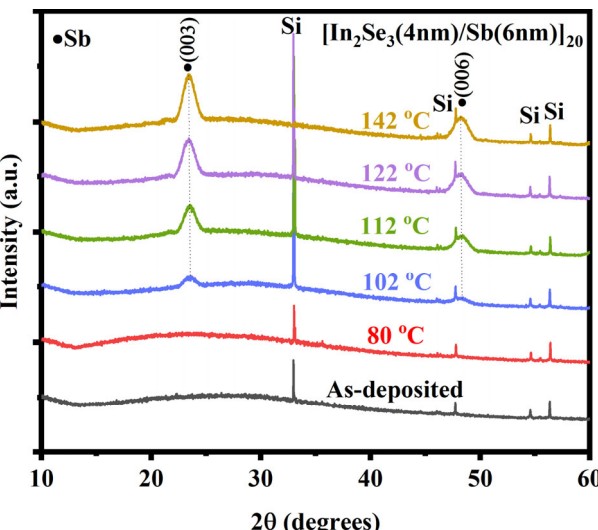

**Figure 4.** XRD spectra of [IS(4 nm)/Sb(6 nm)]$_{20}$ materials at distinct annealing temperatures.

　　　Figure 5 shows the resistance drift curves of the IS/Sb and the Sb films at an isothermal temperature of 60 °C. The relationship between drift resistance and time can be expressed as [5]:

$$R = R_0(t/\tau_0)^{\alpha} \tag{5}$$

where $R$ is the measured resistance at time $t$, $\tau_0$ and $R_0$ are constants describing the initial state of the material, and the exponential factor $\alpha$ is the drift coefficient. The drift index of each film was obtained by linear fitting of the curve. The resistance drift of pure Sb is large (0.1296 ± 0.0009), which is not conducive to the accurate reading of resistance states in Figure 5. When the proportion of the In$_2$Se$_3$ interlayer in the multilayer composite film increases, the drift index of the film declines to 0.0238 ± 0.0005 of [IS(4 nm)/Sb(6 nm)]$_5$, which indicates that the stability of the film resistance is improved [27].

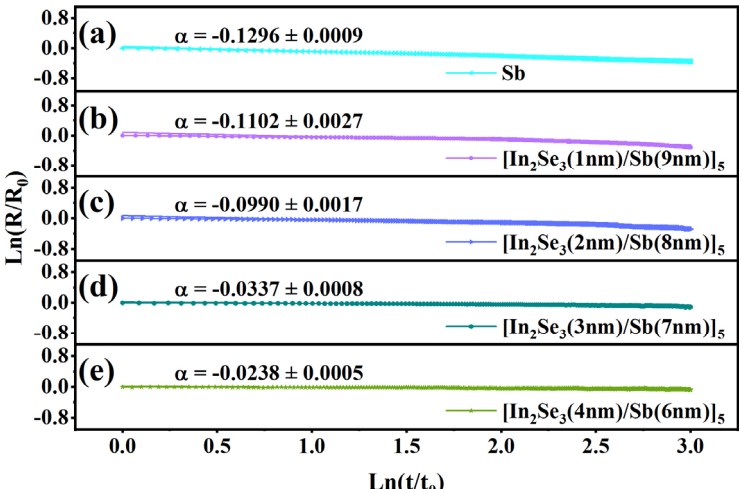

**Figure 5.** (**a**–**e**) are the resistance drift curves of different films.

　　　The photoelectron spectroscopy of the [IS(4 nm)/Sb(6 nm)]$_{20}$ film was tested to analyze the interactions between atoms in the IS/Sb multilayer composite film. Figure 6 shows the full spectrum of the [IS(4 nm)/Sb(6 nm)]$_{20}$ thin film. Figure 6a is a spectral diagram of the Sb 3d orbit. After peak splitting, it can be found that an Sb atom combines with In and O atoms, respectively, to form Sb-O and In-Sb-O bonds. Further, in the 3d spectrogram of the In atoms (Figure 6b), we find that Sb atoms are involved in bonding with In atoms. This indicates that at the interface between the In$_2$Se$_3$ and the Sb film, some In-Se bonds are

broken to form In-Sb bonds, which helps to strengthen the coupling between different film layers and the stability of the interface. The formation of the Sb-O bond is influenced by the active O atoms in the air.

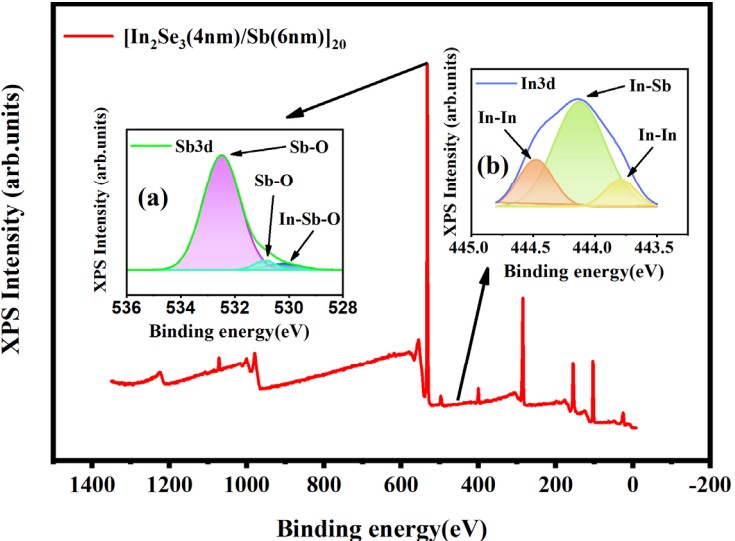

**Figure 6.** XPS spectra of [IS(4 nm)/Sb(6 nm)]$_{20}$ films: (**a**) Sb 3d, (**b**) In 3d.

Figure 7 shows the AFM scanning screen of the [IS(4 nm)/Sb(6 nm)]$_{20}$ film. It is annealed at different temperatures for 5 min. The scanning size is 2000 × 2000 nm. Figure 7a–d show the surface topography of the film under different annealing temperatures, and Figure 7e–h show the bar chart of the surface height of the film. It can be seen that the film surface grain distribution is uniform, with no particularly large protrusions, indicating that the film is relatively dense. Secondly, when the annealing temperature increases, the maximum height of the surface of the film increases from 0.78 nm in the deposited state to 2.45 nm at 142 °C, which shows that the grains in the film grow gradually. The overall grain size is small, which is consistent with the XRD conclusion. Correspondingly, the surface roughness of the film increases from 0.2228 nm in the deposited state to 0.4879 nm at 142 °C, indicating that the surface fluctuation becomes larger due to the growth of the grains. However, the surface is still relatively flat in the crystal state, which is beneficial to maintain the effective contact between the electrode and the film, and helps to improve the reliability of the device [28]. Figure 7i–l show the surface potential of the thin film. The Gauss curvature of the film surface can be expressed as [29]:

$$K_G = \frac{a^2}{\left(a^2 \sin^2 u + b^2 \cos^2 u\right)^2} \tag{6}$$

and the surface charge density of the ellipsoid of rotation is [30]:

$$\sigma = \frac{Q}{4\pi b} \frac{a^2}{\left(a^2 \sin^2 u + b^2 \cos^2 u\right)^{1/2}} \tag{7}$$

For a given ellipsoid $Q$, where $a$ and $b$ are constants, $K_G \propto \sigma^4$ is obtained. The level of potential is highly sensitive to the roughness of the surface. A bigger bump corresponds to a higher potential. Figure 7n shows that when the temperature increases, the electric potential of the film presents an upward trend, indicating that the number of grains increases and the grain size increases.

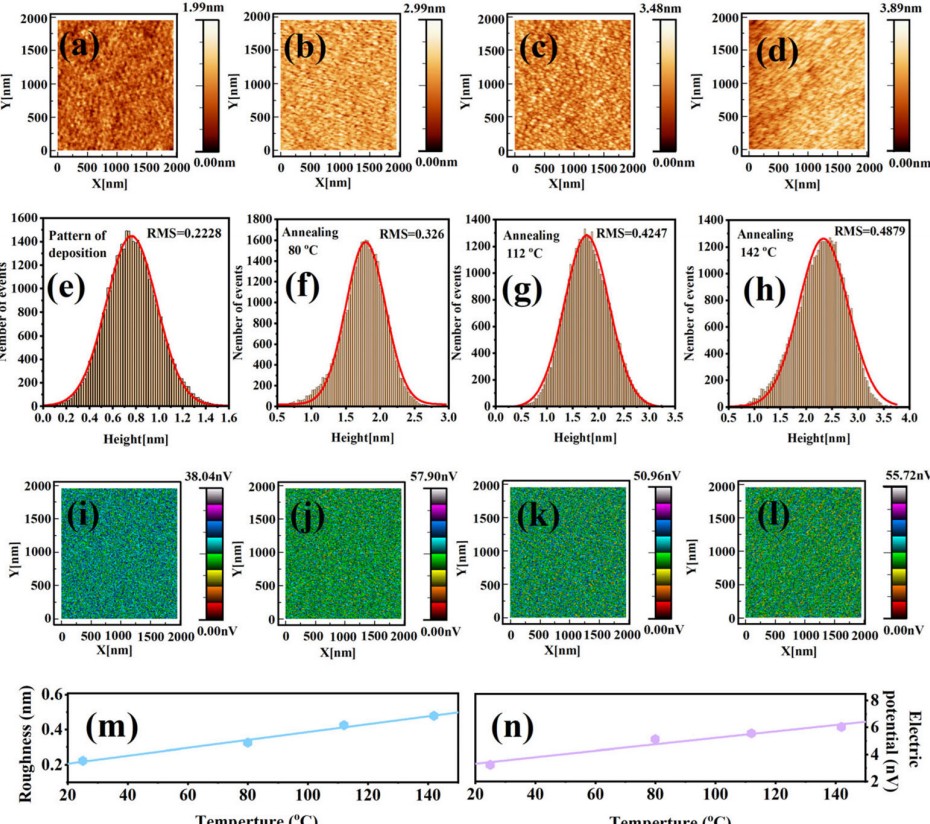

**Figure 7.** (**a–d**) The surface AFM morphologies of [IS(4 nm)/Sb(6 nm)]$_{20}$ materials at distinct annealing temperatures; (**e–h**) Histogram of surface height distribution of thin films at distinct annealing temperatures; (**i–l**) Electrostatic potential of films at distinct annealing temperatures; (**m**) Film surface roughness curve with temperature; (**n**) Temperature variation curve of electrostatic potential on film surface.

Figure 8a shows the fabrication process of the [IS(4 nm)/Sb(6 nm)]$_8$ thin film device. Firstly, a layer of Ag is sputtered on the silicon substrate as the bottom electrode, and then the mask plate is fixed on it. Next, the [IS(4 nm)/Sb(6 nm)]$_8$ film is sputtered on it as the phase change layer, and then the Ag is sputtered on it as the top electrode. Finally, the mask plate is removed. Thus, PCM devices based on [IS(4 nm)/Sb(6 nm)]$_8$ films are prepared. Figure 8b is the image taken during the probe test, and the small illustration at the lower left is the device unit view under the microscope. The illustration in Figure 8c is a schematic diagram of voltage pulse application. Figure 8c shows the *I-V* image of the device. After repeating the experiments many times, it can be seen that the current increases slowly at first as the scanning voltage increases. When the threshold voltage is reached, the voltage suddenly drops, which is a typical negative resistance phenomenon. An ohmic characteristic curve with a high slope is presented, indicating that the deposited film changes into a crystalline state. The [IS(4 nm)/Sb(6 nm)]$_8$ film has a small threshold voltage of 1.26 V, indicating that its Set power consumption is low. Figure 8d shows the *R-V* curves of the [IS(4 nm)/Sb(6 nm)]$_8$ film when a voltage pulse of 1000 ns width is applied. With repeated experiments, it can be observed that the [IS(4 nm)/Sb(6 nm)]$_8$ film can realize the reversible process of Set and Reset. The Reset voltage is between 2–3 V, and the difference between the high and low resistance values is more than two orders of magnitude, which can meet the requirements for the accurate identification of resistance states. This further verifies the practicability of the In$_2$Se$_3$/Sb composite multilayer phase change thin film. In light of the formula for Joule heat: $E_{RESET} = (U_{RESET}^2/R_{RESET}) \cdot t_{RESET}$ [31], the energy consumption for the [IS(4 nm)/Sb(6 nm)]$_8$ film is about $1.72 \times 10^{-13}$ J, while the energy

consumption of the Sb film is $6.53 \times 10^{-13}$ J [4]. Compared with the Sb film, it can effectively reduce energy consumption.

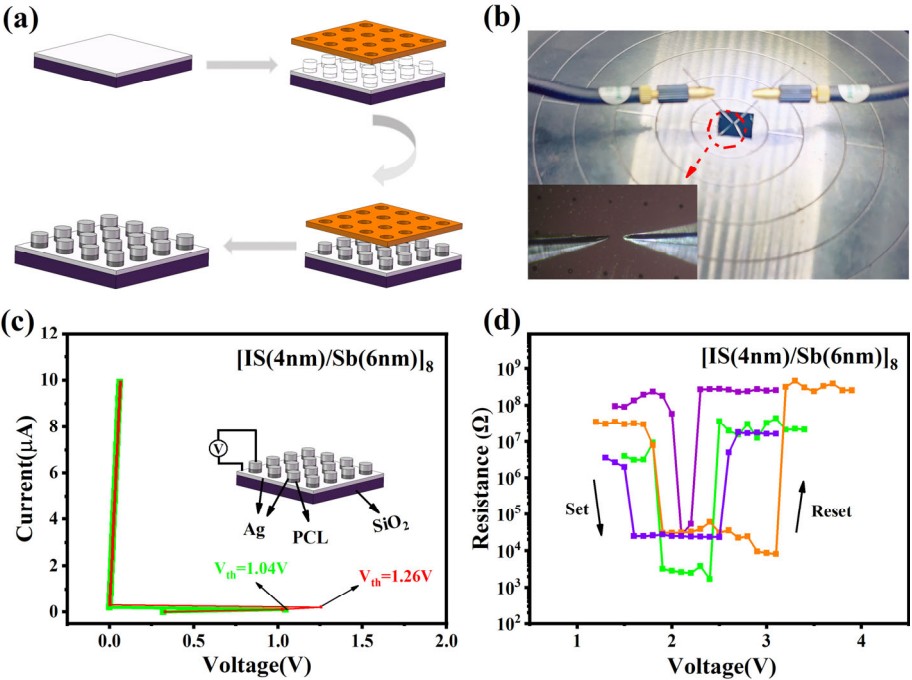

**Figure 8.** (**a**) Preparation process of PCM device of [IS(4 nm)/Sb(6 nm)]$_8$ composite film on silicon wafer as substrate; (**b**) [IS(4 nm)/Sb(6 nm)]$_8$ device test shot; (**c**) The *I-V* curves of the PCM device. Illustrated is a voltage pulse application diagram; (**d**) *R-V* curves of 1000 ns PCM device.

## 4. Implications and Prospects

By adding IS interlayers, the In$_2$Se$_3$/Sb multilayer phase change film was prepared in this study. The thermal stability and resistance drift of phase change thin film are improved. It is of great significance to promote the research and application of phase change memory. In addition, this method also provides an effective means for the development of other phase change films.

## 5. Conclusions

In this paper, the multilayer composite method is adopted by adding an IS interlayer to an Sb film. On the basis of not destroying the original structure of the phase change materials, it gives full play to its original characteristics. At the same time, through the coupling effect between the two kinds of phase change materials, it achieves a synergistic effect, and a complementary advantage effect, so as to improve the performance of the materials. In situ resistance test shows that the $T_c$ of the IS/Sb composite film is 145 °C. It is higher than the 107 °C temperature of pure Sb, which increases its thermal stability. With the increase of the thickness of the IS spacer, the crystallization activation energy of the composite film is increased, which is conducive to the improvement of the reliability of the film. The chemical bonding state, crystal phase structure and microstructure of the materials were studied by means of XPS, XRD and AFM. The R-T and resistance drift curves show that adding an IS spacer can effectively reduce the resistance drift of Sb phase change film. A PCM device based on [IS(4 nm)/Sb(6 nm)]$_8$ film has excellent electrical properties. It reduces power consumption by approximately 74% compared to pure Sb film. In conclusion, by adding an IS spacer, IS/Sb composite film has a higher crystallization temperature and larger crystallization activation energy, which effectively reduces the resistance drift of pure Sb phase change film, and increases its thermal stability. In the future work, we will also study its performance on the flexible substrate, including the influence of bending and compression.

**Author Contributions:** Conceptualization, F.S. and Y.H.; methodology, F.S. and Y.H.; formal analysis, F.S. and Y.H.; investigation, F.S.; writing—original draft preparation, F.S.; writing—review and editing, F.S. and Y.H.; supervision, Y.H., X.Z. and T.L.; funding acquisition, Y.H. All authors have read and agreed to the published version of the manuscript.

**Funding:** This work was supported by National Natural Science Foundation of China (Nos. 11974008, 11774438) and the Opening Project of State Key Laboratory of Silicon Materials (SKL2021-08) and the Engineering Research Center of Digital Imaging and Display, Ministry of Education, Soochow University (SDGC2247).

**Institutional Review Board Statement:** Not applicable.

**Informed Consent Statement:** Not applicable.

**Data Availability Statement:** Not applicable.

**Conflicts of Interest:** The authors declare that they have no known competing financial interests or personal relationships that could have appeared to influence the work reported in this paper.

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
