# Peer review of "Enhancing the Thermal Stability and Reducing the Resistance Drift of Sb Phase Change Films by Adding In2Se3 Interlayers"

_coatings, doi:10.3390/coatings13050927_

Round 1

Reviewer 1 Report

The manuscript entitled “Enhancing the thermal stability and reducing the resistance drift of Sb phase change films by adding In2Se3 interlayers” needs the following amendments before publication.

1.       The novelty of the presented work needs to be highlighted in the abstract, the last passage of the introduction, and the conclusion.

2.       The underneath mechanism of higher crystallization temperature, larger crystallization activation energy, smaller resistance drift and better thermal stability of the composite multilayer In2Se3/Sb films as compared to pure Sb films must be highlighted in the revised manuscript.

3.       The abstract and the conclusion sections must be supplemented with the scientific findings of the work.

4.       The statements “Compared with pure Sb films, the composite multilayer In2Se3/Sb films had higher crystallization temperature, larger crystallization activation energy, smaller resistance drift and better thermal stability.” and “Compared with Sb films, [In2Se3(4nm)/Sb(6nm)]8 films had smaller threshold voltage and lower power consumption. In conclusion, the In2Se3 interlayers effectively inhibited the resistance drift of phase change films and enhanced the thermal stability.” must be quantified.

5.       A standard referencing style is to be followed, especially for the introduction section.

6.       Lumped references should be avoided, and individual contributions should be detailed.

7.       The introduction section is too short, and the relevant published literature is not sufficiently explored. It is suggested to extend this section by citing more relevant published literature and establish a research gap to be filled by this work.

8.       A list of acronyms should be added to define all the abbreviations and equation parameters used in the manuscript.

9.       The statement on lines 235-236 “In situ resistance test indicates that the Tc of IS/Sb film is 145 ℃, which is more high than that of pure Sb at 107 ℃.” should be corrected grammatically. ‘more high’ may be replaced with 'higher’ and % higher will be more effective to quantify the increase.

10.   The statement (lines236-237) “The crystallization curves of films with distinct proportion components all show a slow and then steep decline.” must be backed by the reasons for this slow and steep decline.

11.   A new section ‘Implications and prospects’ and the possible future works should be explained to overcome these limitations.

12.   Conclusion section should be rewritten to highlight the novelty, a brief methodology, and the facts and figures of the main findings. 

Authors are suggested to improve the English grammar and typos in the revised manuscript. 

Author Response

Dear editor and reviewers:

Thank you for your letter and for the reviewers’ comments concerning our manuscript entitled “Enhancing the thermal stability and reducing the resistance drift of Sb phase change films by adding In2Se3 interlayers” (ID: coatings-2365995).

Those comments are all valuable and very helpful for revising and improving our paper, as well as the important guiding to our researches. We have studied comments carefully and have made correction which we hope meet with approval. Revised portions are highlighted in yellow in the paper. The main corrections in the paper and the responds to the reviewer’s comments are as following.

Once again, we acknowledge your significant comments and constructive suggestions very much, which are valuable in improving the quality of our manuscript.

Yours sincerely,

Feng Su, Yifeng Hu

May1, 2023

Responds to the reviewer 1 comments

1.Question 1:The novelty of the presented work needs to be highlighted in the abstract, the last passage of the introduction, and the conclusion.

Response: Thanks for the reviewer's suggestions. We have revised the abstract, introduction and conclusion. (See the highlighted portion in the article)

2.Question 2:The underneath mechanism of higher crystallization temperature, larger crystallization activation energy, smaller resistance drift and better thermal stability of the composite multilayer In2Se3/Sb films as compared to pure Sb films must be highlighted in the revised manuscript.

Response: Thanks to the reviewers for the valuable comments. The better thermal stability of In2Se3/Sb films is mainly due to the addition of In2Se3 layer with higher phase transition temperature and the clamping effect of multilayer interface on Sb layer, which reduces the phase transition region of Sb and inhibits its phase transition, thus leading to better thermal stability. (See the highlighted portion in this article for details)

3.Question 3:The abstract and the conclusion sections must be supplemented with the scientific findings of the work.

Response: Thanks to the valuable comments of reviewers. We have improved and supplemented the abstract and conclusion of the manuscript.

4.Question 4:The statements “Compared with pure Sb films, the composite multilayer In2Se3/Sb films had higher crystallization temperature, larger crystallization activation energy, smaller resistance drift and better thermal stability.” and “Compared with Sb films, [In2Se3(4nm)/Sb(6nm)]8 films had smaller threshold voltage and lower power consumption. In conclusion, the In2Se3 interlayers effectively inhibited the resistance drift of phase change films and enhanced the thermal stability.” must be quantified.

Response: Thanks for the reviewer's comments. We have added the quantitative descriptions of phase transition temperature, crystallization activation energy, resistance drift and power consumption in the paper.

5.Question 5:A standard referencing style is to be followed, especially for the introduction section.

Response: Thank you very much for the reviewer's reminding. We have modified and unified the reference style of the literature.

6.Question 6:Lumped references should be avoided, and individual contributions should be detailed.

Response: Thanks for the reviewer's suggestion. We have dispersed the literature citations to make them more scientific.

7.Question 7:The introduction section is too short, and the relevant published literature is not sufficiently explored. It is suggested to extend this section by citing more relevant published literature and establish a research gap to be filled by this work.

Response: Thanks for the reviewer's reminding. We have added the explanation of the research significance of this paper in the introduction.

8.Question 8:A list of acronyms should be added to define all the abbreviations and equation parameters used in the manuscript.

Response: Thanks to the reviewer's suggestion. We have checked every acronym and equation parameter in the manuscript and added specific full names and explanations where they first appear.

9.Question 9:The statement on lines 235-236 “In situ resistance test indicates that the Tc of IS/Sb film is 145 ℃, which is more high than that of pure Sb at 107 ℃.” should be corrected grammatically. ‘more high’ may be replaced with 'higher’ and % higher will be more effective to quantify the increase.

Response: Thank you very much for the reviewer's valuable suggestions. We have corrected the grammatical errors you pointed out and made significant modifications to the conclusion according to your suggestions.

10.Question 10:The statement (lines236-237) “The crystallization curves of films with distinct proportion components all show a slow and then steep decline.” must be backed by the reasons for this slow and steep decline.

Response: Thanks for the reviewer's suggestion. Because the crystallization process of the film is divided into four stages, namely incubation period, nucleation period, growth period and grain coarse period, so the crystallization curve of the film presents a slow and then steep decline process. We have added this part of the explanation to the text.

11.Question 11:A new section ‘Implications and prospects’ and the possible future works should be explained to overcome these limitations.

Response: Thanks to the reviewer's suggestion. We have added the prospect of this work in the introduction part and the plan for further research in the conclusion part.

12.Question 12:Conclusion section should be rewritten to highlight the novelty, a brief methodology, and the facts and figures of the main findings.

Response: Thanks very much for the reviewer's suggestion. We have made significant modifications and improvements in the conclusion.

Reviewer 2 Report

This paper presents a novel development of pure Sb films and composite multilayer In2Se3/Sb thin films on SiO2/Si substrates. Resistance, absorbance and other optical aspects have been presented. Graphs are of very good quality. Discussion is also thorough and results are plausible. I recommend acceptance of paper in its current form.

Author Response

Thank you very much for your recognition and support.

Yours sincerely,

Feng Su, Yifeng Hu

May1, 2023

Reviewer 3 Report

The manuscript titled by “Enhancing the thermal stability and reducing the resistance drift of Sb phase change films by adding In2Se3 interlayers” reports a comprehensive study of the physical and electrical properties of phase change thin films, the pure Sb films and composite multilayer In2Se3/Sb thin films prepared on SiO2/Si substrate. Based on resistivity, absorption, crystallization, AFM, XRD, XPS measurement, the authors report that the composite multilayer In2Se3/Sb films had higher crystallization temperature, larger crystallization activation energy, smaller resistance drift and better thermal stability. Finally, the authors show that PCM devices based on [IS(4nm)/Sb(6nm)]8 films have good electrical performance and low threshold voltage.

The experiments are well planned and carried out. The paper is well written. Although there are some issues about the data analysis and arguments, this work is overall interesting and timely. the paper can be published after the authors revise their paper by considering the following comments.   

11  The crystallization-time curves and crystallization activation energy of the films were measured in Fig. 3. How does the authors carry out this work?

  2 The reflectance spectrum of the thin film was observed by near infrared spectrophotometer with wavelength range of 400-2500 nm. Where is the data with wavelength range of 400-2500 nm?

  3 Figure 2 shows Kubelka-Munk function curves of [IS(4nm)/Sb(6nm)]20 film at distinct annealing temperatures. Fig. 2 shows that Eg gradually decreases from 0.935 eV~25 ℃ to 0.887 116 eV~102 ℃, 0.87 eV~122 ℃ and 0.861 eV~180 ℃. How do the authors obtain these gaps, what is the meanings of these gaps? And what is the data for other films with ifferent components? In Fig. 2, the base line zero is too far away from the data, this makes the analysis unreasonable.

   4 In Figure 3 (e)-(h), it really makes no sense to fit the data with linear function, are these values of n really meaningful?  

   5 Do XPS spectra of [IS(4nm)/Sb(6nm)]20 films not show any In-Se binging?

   6  The data presentation in Fig. 8(c) is very poor. It must be improved.

Author Response

Dear editor and reviewers:

Thank you for your letter and for the reviewers’ comments concerning our manuscript entitled “Enhancing the thermal stability and reducing the resistance drift of Sb phase change films by adding In2Se3 interlayers” (ID: coatings-2365995).

Those comments are all valuable and very helpful for revising and improving our paper, as well as the important guiding to our researches. We have studied comments carefully and have made correction which we hope meet with approval. Revised portions are highlighted in yellow in the paper. The main corrections in the paper and the responds to the reviewer’s comments are as following.

Once again, we acknowledge your significant comments and constructive suggestions very much, which are valuable in improving the quality of our manuscript.

Yours sincerely,

Feng Su, Yifeng Hu

May1, 2023

Responds to Reviewer 3 Comments

1.Question 1:The crystallization-time curves and crystallization activation energy of the films were measured in Fig. 3. How does the authors carry out this work?

Response: Thanks for the reviewer's questions. As shown in Fig.3 a-d, they show the crystallization proportion-time curves of thin films of different components under isothermal annealing, while e-h show the curves of Ln[-ln(1-χ)]-Ln(t). Using a combination of a Keithley 6517B high resistance meter and a Linkam HFS600E-PB2 hot and cold table, we measured the resistance and time of the samples at four isothermal temperatures. We then normalize the resistance and define the failure time when the resistance drops to 50% of the initial value. The crystallization activation energy of different films was calculated by Arrhenius equation. The Avrami index of thin films was calculated by Avrami equation and the crystallization mechanism of thin films was evaluated.

2.Question 2:The reflectance spectrum of the thin film was observed by near infrared spectrophotometer with wavelength range of 400-2500 nm. Where is the data with wavelength range of 400-2500 nm?

Response: Thanks for the questions raised by the reviewer. We tested the reflection spectrum in the range of 400~2500 nm (not given), and selected the data in the range of 902~1016 nm for research, that is, the curve corresponding to the horizontal coordinate of 1.22~1.38 eV in the figure.

3.Question 3:Figure 2 shows Kubelka-Munk function curves of [IS(4nm)/Sb(6nm)]20 film at distinct annealing temperatures. Fig. 2 shows that Eg gradually decreases from 0.935 eV~25 ℃ to 0.887 116 eV~102 ℃, 0.87 eV~122 ℃ and 0.861 eV~180 ℃. How do the authors obtain these gaps, what is the meanings of these gaps? And what is the data for other films with different components? In Fig. 2, the base line zero is too far away from the data, this makes the analysis unreasonable.

Response: Many thanks to the reviewers for their valuable comments.

    We process the reflectance~wavelength data measured by near infrared spectrophotometer. It can be calculated by the Kubelka-Munk formula (1-R)2/2R=K/S , where K, S and R are absorption coefficient, scattering coefficient and reflectance respectively. As shown in the figure, the intersection of the line and X-axis obtained by processing and fitting is the band gap. The carrier density in semiconductor is proportional to [-Eg/(2KT)], and the decrease of band gap will lead to the increase of carriers, which is the main reason why the resistivity of the film decreases with the increase of annealing temperature. It is consistent with the R-T curve in Figure 1a.

During the phase transition, the energy band of the film changes significantly with the heating temperature. Therefore, the energy bands of [IS(4nm)/Sb(6nm)]20 at different temperatures were selected for study in this paper. The results show that the energy band value decreases with the increase of temperature. As for the energy bands of other films, their variation with temperature is similar to [IS(4nm)/Sb(6nm)]20, so the results are not shown here.

In order to more clearly display the band gap value of the film studied in this paper, we set the horizontal coordinate range (0.6~1.6 eV) (as shown in the figure) to better display the difference of this part of band gap value, which will not affect the accuracy of Eg value.

4.Question 4:In Figure 3 (e)-(h), it really makes no sense to fit the data with linear function, are these values of n really meaningful? 

Response: Thanks for the reviewer's comments. According to Avrami equation, χ(t)=1-exp[-(Kt)n], where Avrami exponent n is related to the phase transition mechanism and depends on the decay of the nucleation rate. K is a constant and t is time. The values of n represent different crystallization processes in different ranges. In this study, the values of n are all less than 2.5, indicating that the crystallization mechanism belongs to one-dimensional growth. This mechanism usually corresponds to a faster phase transition rate, which may be one of the reasons affecting the phase transition rate of thin films.

5.Question 5:Do XPS spectra of [IS(4nm)/Sb(6nm)]20 films not show any In-Se binging?

Response: Thanks to the reviewer for your question. XPS spectrum includes In-Se combination. Since we studied multilayer composite films by measuring XPS spectrum, our research mainly focused on whether the film formed new chemical bonds after composite, that is, whether there was a bond between In and Sb, so we did not pay too much attention to the bond between In and Se.

6.Question 6:The data presentation in Fig. 8(c) is very poor. It must be improved.

Response: Thank you for the questions pointed out by the reviewer. I am very sorry for the presentation form of Fig. 8c. Due to the good repeatability of the amorphous to crystalline transition of the film, the two I-V curves overlap together, which brings you a bad review experience. We have made appropriate improvements to the diagram.

Reviewer 4 Report

The paper is devoted to the thermal behavior of thin Sb film with addition of In2Sb3 interlayers. The work sounds scientific. The only question regarding the results is why authors do not presented any theoretical calculations regarding band gaps and energy. I think the work after corrections can be published.

Minor issues:

". The reflectance spectrum of the thin film was observed 63 by near infrared spectrophotometer with wavelength range of 400-2500 nm.". The reflectance spectrum of the thin film was observed 63 by near infrared spectrophotometer with wavelength range of 400-2500 nm. please, specify the instrument

"X-ray diffractometer (XRD) was used to test the X-ray lines of the film between 10o and 60o" please, specify the instrument and the same for AFM.

The pressure in the vacuum chamber is not low. Please specify the composition of residual gases in the chamber (e.g. mass spectrometry). 

How did you control the sample temperature? How did you calibrate temperature or it was PT100?

How did you measure the layer resistance and by what device?

Please improve the quality of figure 3.

What do you mean by "multiple interfaces" L 139 - it is related to the crystallizes or to crystals facets with different orientation or facets with different composition? High indexed facets have lower energy and crystals on them growths faster. Maybe it is a poisoning effect of the crystal growth by other compound?

Author Response

Dear editor and reviewers:

Thank you for your letter and for the reviewers’ comments concerning our manuscript entitled “Enhancing the thermal stability and reducing the resistance drift of Sb phase change films by adding In2Se3 interlayers” (ID: coatings-2365995).

Those comments are all valuable and very helpful for revising and improving our paper, as well as the important guiding to our researches. We have studied comments carefully and have made correction which we hope meet with approval. Revised portions are highlighted in yellow in the paper. The main corrections in the paper and the responds to the reviewer’s comments are as following.

Once again, we acknowledge your significant comments and constructive suggestions very much, which are valuable in improving the quality of our manuscript.

Yours sincerely,

Feng Su, Yifeng Hu

May 1, 2023

Responds to the reviewer 4 comments

1.Question 1:The reflectance spectrum of the thin film was observed by near infrared spectrophotometer with wavelength range of 400-2500 nm.". The reflectance spectrum of the thin film was observed by near infrared spectrophotometer with wavelength range of 400-2500 nm. please, specify the instrument.

Response: Thanks for the question raised by the reviewer. The instrument we used to observe the reflection spectrum of the thin film is the 7100CRT near infrared spectrophotometer.

2.Question 2:X-ray diffractometer (XRD) was used to test the X-ray lines of the film between 10o and 60o" please, specify the instrument and the same for AFM.

Response: Thanks for the reviewer's questions. The instrument we tested for XRD was XPert POWDER of Panaco Company, and the AFM instrument we tested was FM-Nanoview1000 AFM.

3.Question 3:The pressure in the vacuum chamber is not low. Please specify the composition of residual gases in the chamber (e.g. mass spectrometry).

Response: Thank you very much for your questions. The pressure in the vacuum chamber is lower than 4×10-4 Pa, which is in a high vacuum degree. There might be a tiny amount of air there, but the effect on our experiment is negligible. In the process of the experiment, argon gas was also introduced for sputtering, and the specific composition of the residual gas was not further explored. However, we will carry out relevant experiments to study in the follow-up work.

4.Question 4:How did you control the sample temperature? How did you calibrate temperature or it was PT100?

Response: Thank you for your questions. We tested the resistance-temperature curve by combining the Keithley 6517B high resistance meter with the Linkam HFS600E-PB2 cold and hot station. Test only need to put the sample on the hot and cold table and cover the utensils, through the computer control of the rise and fall of the hot and cold table. Before the experiment, PT100 temperature sensor will be used to calibrate the temperature.

5.Question 5:How did you measure the layer resistance and by what device?

Response: Thanks for the reviewer's question. We measured the resistance of the film through the Keithley 6517B high resistance meter and the Linkam HFS600E-PB2 cold and hot table. What is measured in this paper is the overall resistance of the film, not the resistance of individual layers. We used a digital source meter 2400 for device resistance test.

6.Question 6:Please improve the quality of figure 3.

Response: Thank you very much for your reminding. The annotation in Figure 3 is a little vague. We have tried our best to present the content in the picture more clearly.

7.Question 7:What do you mean by "multiple interfaces" L 139 - it is related to the crystallizes or to crystals facets with different orientation or facets with different composition? High indexed facets have lower energy and crystals on them growths faster. Maybe it is a poisoning effect of the crystal growth by other compound?

Response: Thank you very much for your questions. As our film is composite multilayer, IS and Sb are alternately sputtered on Si substrate during preparation, so for the composite film, it will have obvious layered structure, while pure Sb film does not have such layered structure. Because of the existence of multiple interfaces in the composite film, the crystallization of Sb is inhibited, which makes the grains not easy to grow up. Different crystallization index corresponds to different crystallization mechanism. When n value is less than 2.5, it indicates that the crystallization mechanism belongs to one-dimensional growth. It has to do with the composition of the film. This mechanism usually corresponds to a faster phase transition rate, which may be one of the reasons affecting the phase transition rate of thin films.

Round 2

Reviewer 1 Report

Authors are suggested to add the proposed section before the conclusion instead of adding the text to the introduction section. 

Moderate English editing is required

Author Response

Dear editor and reviewers:

Thank you for your letter and for the reviewers’ comments concerning our manuscript entitled “Enhancing the thermal stability and reducing the resistance drift of Sb phase change films by adding In2Se3 interlayers” (ID: coatings-2365995).

Those comments are all valuable and very helpful for revising and improving our paper, as well as the important guiding to our researches. We have studied comments carefully and have made correction which we hope meet with approval. Revised portions are highlighted in yellow in the paper. The main corrections in the paper and the responds to the reviewer’s comments are as following.

Once again, we acknowledge your significant comments and constructive suggestions very much, which are valuable in improving the quality of our manuscript.

Yours sincerely,

Feng Su, Yifeng Hu

May 7, 2023

Responds to the reviewer 1 comments

  1. Comments and Suggestions for Authors:Authors are suggested to add the proposed section before the conclusion instead of adding the text to the introduction section.

Response: Thank you for your patient suggestion. We have added the implications and prospects before the conclusion.

      2.Comments on the Quality of English Language:Moderate                 English editing is required.

Response: Thanks to the reviewer's careful review and reminder. We have re-checked the article and improved the English.

Reviewer 3 Report

The authors have responded my comments. I am satisfied with most of their responses except for the  last figures. Anyway, what the results show is there, I think the paper can be published as is.

Author Response

         Thank the reviewers for their patience and valuable comments on our manuscript. We will pay attention to avoid these problems in our future work. Thank you very much.

Yours sincerely,

Feng Su, Yifeng Hu

May 7, 2023